# Mortality Associated with Idiopathic Pulmonary Fibrosis in Northeastern Italy, 2008–2020: A Multiple Cause of Death Analysis

**DOI:** 10.3390/ijerph18147249

**Published:** 2021-07-06

**Authors:** Alessandro Marcon, Elena Schievano, Ugo Fedeli

**Affiliations:** 1Unit of Epidemiology and Medical Statistics, Department of Diagnostics and Public Health, University of Verona, Strada Le Grazie 8, 37134 Verona, Italy; 2Epidemiological Department, Azienda Zero Regione del Veneto, via Jacopo Avanzo 35, 35132 Padova, Italy; elena.schievano@azero.veneto.it (E.S.); ugo.fedeli@azero.veneto.it (U.F.)

**Keywords:** idiopathic pulmonary fibrosis, mortality, time trends analysis, multiple cause of death, underlying cause of death, COVID-19, epidemiology

## Abstract

Mortality from idiopathic pulmonary fibrosis (IPF) is increasing in most European countries, but there are no data for Italy. We analysed the registry data from a region in northeastern Italy to assess the trends in IPF-related mortality during 2008–2019, to compare results of underlying vs. multiple cause of death analyses, and to describe the impact of the COVID-19 epidemic in 2020. We identified IPF (ICD-10 code J84.1) among the causes of death registered in 557,932 certificates in the Veneto region. We assessed time trends in annual age-standardized mortality rates by gender and age (40–74, 75–84, and ≥85 years). IPF was the underlying cause of 1310 deaths in the 2251 certificates mentioning IPF. For all age groups combined, the age-standardized mortality rate from IPF identified as the underlying cause of death was close to the European median (males and females: 3.1 and 1.3 per 100,000/year, respectively). During 2008–2019, mortality rates increased in men aged ≥85 years (annual percent change of 6.5%, 95% CI: 2.0, 11.2%), but not among women or for the younger age groups. A 72% excess of IPF-related deaths was registered in March–April 2020 (mortality ratio 1.72, 95% CI: 1.29, 2.24). IPF mortality was increasing among older men in northeastern Italy. The burden of IPF was heavier than assessed by routine statistics, since less than two out of three IPF-related deaths were directly attributed to this condition. COVID-19 was accompanied by a remarkable increase in IPF-related mortality.

## 1. Introduction

Idiopathic pulmonary fibrosis (IPF) is a chronic, progressive, and fibrotic lung disease of unknown causes, occurring primarily in older adults [1]. It is the most common idiopathic interstitial pneumonia and holds the worst prognosis [2]. Evidence suggests that the incidence and prevalence of IPF range between 2–30 and 10–60 per 100,000/year, respectively, but comparisons across studies are difficult owing to differences in data sources and case definitions [3]. In patients with IPF, healthy lung tissue is replaced by extracellular matrix and the alveolar structure is destroyed, which leads to disrupted gas exchange and ultimately respiratory failure and death [4]. The median survival time from diagnosis is 2–4 years [5,6]. According to the World Health Organization mortality database, the median age-standardized mortality rates from IPF in European countries in 2001–2013 were 3.75 per 100,000 and 1.50 per 100,000 for males and females, respectively [7]. There are substantial differences in IPF mortality rates between EU member states, with an overall increase in mortality in most countries but a decrease observed in others, and a persistent sex-mortality gap [7,8]. To date, no data on mortality are available for Italy.

Most of the evidence available on IPF-related mortality is based on standard statistics limited to the underlying cause of death (UCOD) derived from the set of causes reported on death certificates. The UCOD approach is thought to underestimate mortality due to diseases that contribute to death without being selected as the underlying cause [9]. Multiple cause of death (MCOD) analyses for IPF, which search for the disease listed anywhere on a death certificate, are warranted to assess the full burden of IPF-related mortality, including the impact of the new Coronavirus disease (COVID-19) pandemic. Indeed, interstitial lung diseases seem to increase susceptibility to, and the severity of, COVID-19 [10]. International comparisons of mortality rates and time trends are now feasible based on the MCOD approach, since most countries have adopted a software-guided standardized selection of the UCOD from mortality records.

MCOD analyses for IPF, which, to date, are available only for the United States (US), Australia, England, and Wales, revealed considerably more deaths compared to the UCOD analyses [11]. The US is the only country where MCOD for IPF have been extensively analysed [2,12]. Mortality increased until 2004 and declined thereafter, especially in younger age classes; improvements in the management of IPF and comorbidities might have contributed to the change in time trends observed in the US in the more recent period [13,14]. Outside the US, there is a lack of MCOD data for IPF to assess time trends and comorbidities contributing to death.

The aims of the present report are to investigate mortality associated with IPF in northeastern Italy from 2008 to 2019, to compare results from the UCOD and MCOD analyses, and to provide preliminary data on the impact of the first COVID-19 surge in March–April 2020.

## 2. Methods

Copies of the death certificates of residents in the Veneto region (northeastern Italy, about 4.9 million population) are transmitted to the Regional Epidemiological Department for coding of the causes of death according to the International Classification of Diseases, 10th Edition (ICD-10). Standard mortality statistics are based on the UCOD, identified from all the conditions reported in a certificate according to internationally adopted rules. For the period 2008–2017, such rules have been applied by means of the Automated Classification of Medical Entities, a program developed by the US National Centre for Health Statistics to standardize assignment of the UCOD [15]. Starting from 2018, the selection of the UCOD has been performed by means of the IRIS software, currently adopted in most European countries [16].

All death certificates from 1 January 2008 to 31 December 2019 that reported the ICD-10 code J84.1 (other interstitial pulmonary diseases with fibrosis) were selected. All the analyses were stratified by gender a priori. The number of deaths from IPF according to the UCOD and MCOD, age-specific mortality rates, and the proportional mortality (the share of IPF-related deaths out of all deaths) were computed. Annual age-standardized mortality rates (direct standardization, European standard population, 2013) based both on the UCOD and on the MCOD were estimated to explore temporal trends separately for subjects aged <75, 75–84, and ≥85 years.

Differences in time trends in mortality between age groups were tested using likelihood ratio tests comparing Poisson regression models with vs. without interaction terms between year (centered at 2008) and age group (40–74, 75–84, ≥85 years). The annual percentage changes in mortality rates through the study period, with 95% confidence intervals (95% CI), were then estimated separately for each age group using Poisson regression models with the year as the independent variable, adjusting for age coded in 5-years classes (to control for residual confounding within strata). All the models included an offset for log person-years.

The distribution of the most common UCOD in certificates with any mention of IPF was investigated. Associations between IPF and selected complications/comorbidities in death certificates were expressed as adjusted odds ratios (OR) with 95% CIs estimated through conditional logistic regression; each model included one complication/comorbidity (present vs. absent) as the dependent variable, IPF (mentioned in any part of a death certificate vs. not mentioned) as the independent variable, and all combinations between gender, age class, and the year of death as strata.

As in previous reports from the US [14], a sensitivity analysis on the temporal trends of IPF mortality was carried out after excluding death certificates with mentions of conditions that are causally related to pulmonary fibrosis, namely connective tissue diseases (ICD-10 codes M05–M08, M32–M36), sarcoidosis (D86), pneumoconiosis, hypersensitivity pneumonitis, and radiation fibrosis (J60–J65, J67, J70.1).

Lastly, the monthly count of deaths with mentions of IPF was compared between the first semester of 2020 (only provisional data available) and the previous years (2008–2019), to assess the impact of the COVID-19 epidemic on IPF-related mortality. The ratio of IPF-related deaths observed in March–April 2020 (corresponding to the first epidemic wave in the Veneto region) to those expected (the average of the 2008–2019 period) was computed with 95% CIs based on the Poisson distribution. The statistical analyses were conducted using STATA software, release 16 (StataCorp, College Station, TX, USA).

## 3. Results

Out of the 557,932 overall deaths registered in 2008–2019, IPF was selected as the UCOD in 1310, and mentioned anywhere in the certificate in 2251 (1366 males and 885 females), corresponding to a proportional mortality of 0.4% (0.5% and 0.3% among males and females, respectively) (Table 1). For both the UCOD and the MCOD analyses, IPF-related mortality rates were higher in males compared to females, and they increased steeply with age. In the MCOD analysis, mortality rates peaked at 65.2 and 28.1 per 100,000/year for males and females aged 85–89 years, respectively (Table 1).

IPF was selected as the UCOD in 58.2% of all the certificates with any mention of the disease; further common UCOD were ischemic heart diseases (6.6%), other circulatory diseases (8.3%), COPD (3.9%), lung cancer (3.9%), and other neoplasms (4.6%) (Table 2). Frequently reported complications/comorbidities in death certificates mentioning IPF were influenza and pneumonia (15.3%), COPD (12.9%), and pulmonary hypertension (12.5%) (Table 3). The latter was also the condition most strongly related to a mention of IPF in death certificates. In fact, pulmonary hypertension was 16-times more likely to be reported in death certificates mentioning IPF, compared to certificates with no mention of IPF (OR 15.7, 95% CI: 13.8, 17.9) (Table 3). The second strongest association was found for the conditions causally related to pulmonary fibrosis (OR 9.8, 95% CI: 8.3, 11.5), which were mentioned in 8.0% of IPF-related death certificates.

During 2008–2019, age-standardized mortality rates based on the UCOD were 3.1 per 100,000/year among males and 1.3 per 100,000/year among females, which are both slightly lower than the European median rates for the period 2001–2013 [7]; such figures increased to 5.3 and 2.2 per 100,000/year, respectively, in the MCOD analyses.

When looking at all ages combined, there was an interaction between calendar year and age group, limited to males (p_interaction_ = 0.046 for UCOD and 0.063 for MCOD analyses, respectively). Indeed, a significant increasing trend in IPF-related mortality was evident among males aged ≥85 years, but not for the other age groups (Figure 1): the annual percent change was 6.5% (95% CI: 2.0, 11.2%) in the UCOD analysis and 3.5% (95% CI: 0.2, 6.9%) in the MCOD analysis (Appendix A). In this group, estimated annual percent changes were very similar when excluding the death certificates mentioning known causes of pulmonary fibrosis (UCOD: 7.0%, 95% CI: 2.4, 11.7%; and MCOD: 3.7%, 95% CI: 0.3, 7.2%). No significant time trend in IPF-related mortality was seen for females (Figure 2 and Appendix A).

In 2008–2019, a strong seasonality was evident, with IPF-related deaths peaking in January and reaching a minimum in August (mean monthly counts of 20 and 12, respectively) (Figure 3). According to the provisional data, a 72% excess of IPF-related deaths was registered in March–April 2020 (54 deaths observed vs. 31 expected; mortality ratio 1.72, 95% CI: 1.29, 2.24), corresponding to the first COVID-19 surge in the Veneto region [17]. The excess death count was also evident when comparing 2020 with single years in 2008–2019 (Appendix A).

## 4. Discussion

In this analysis of data from the Veneto region, a large administrative area in northern Italy, we found that mortality from IPF identified as the UCOD was close to the European median. An increase in IPF-related mortality between 2008 and 2019 in men aged 85 years and older was evident both when IPF was assigned as the UCOD and when it was mentioned in death certificates as one of the causes of death. IPF-mortality rates were stable among women throughout the period. Preliminary data suggest that the first surge of COVID-19, which struck harshly in northern Italy before the other European regions during early 2020, was accompanied by a remarkable increase in IPF-related mortality.

To our knowledge, this is the first report on IPF-related mortality using MCOD registry data from continental Europe. In our study, IPF-related deaths increased by about 65% when considering any mention of the disease with respect to standard analyses limited to the UCOD. This percentage is consistent with mortality studies carried out in England and Wales (2001–2012, +73%), and the US (1999–2010, +59%), but lower than in Australia (2000–2011, +91%) [11]. However, we acknowledge that the comparison of mortality data across countries is made difficult by a variable degree of ascertainment of IPF in real-life practice [7,8]. In fact, distinguishing IPF from other interstitial lung diseases requires a combination of clinical, radiological, and histological findings, and the availability of expertise and diagnostic equipment may still be suboptimal in some countries one decade after the 2011 consensus statement on IPF diagnostic criteria [18]. Moreover, there is no mandatory registration of IPF in any country worldwide and several national registries still report only the most common respiratory causes of death, such as pneumonia or COPD [8,19]. An analysis of causes of death in a UK cohort of patients with IPF suggested that mortality data may underreport IPF diagnosis by 20–30% [11].

We documented higher IPF-related mortality in males and older subjects compared to their counterparts [11,12,19]. These findings are in agreement with the higher prevalence and incidence of the disease that has been observed for the same demographic groups in Italy and elsewhere [5,20], but also with the greater frequency of comorbidities in older men, which could contribute to mortality among patients with IPF.

To date, most of the evidence is consistent with a secular increase in IPF-related mortality globally [2,7,8,11,19,21], with the previously mentioned exception for the US [13,14]. Our study, the first documenting trends of IPF-related deaths in Italy, suggested an increase in mortality among men but not among women. Indeed, the sex-mortality gap appears to be increasing over time worldwide, although some countries further highlighted a tendency for an increase in women [7].

In the UCOD analysis, there was a marked increase among men aged 85 or older, as shown by the highest annual percent increase in the group characterized by the highest absolute mortality rates. As reported by others, the rapid nature of such an upward trend suggests that an increase in disease ascertainment may play a dominant role [19]. Among older men, however, mortality with IPF also showed an increase (MCOD analysis), which could also reflect increased mortality from comorbidities that are common in this population group.

Several mechanisms have been advocated to explain the rise in IPF-related mortality worldwide [7], which could also be involved in the trends in male mortality observed in our study. First, increasing mortality may be a true biological phenomenon due to the ageing of the Italian population. Second, it could be partly due to a long-term trend for improved recognition in IPF in clinical practice, along with the increased availability of diagnostic equipment (high-resolution CT scanners) and the growing expertise of multidisciplinary teams of clinicians with specific experience in interstitial lung diseases [7]. The update of IPF diagnostic criteria in the consensus document in 2011 may have also contributed to changing mortality trends. However, none of the advocated mechanisms can concisely explain the gender-gap in time trends observed in our study.

This report confirms findings available only from the US that cardiovascular diseases (especially ischemic heart diseases), COPD, and neoplasms (mainly lung cancer) are the most common causes of death when IPF is mentioned in death certificates but not selected as the UCOD [2,14].

The MCOD approach allowed us to investigate the association between IPF and complications increasing the risk of death, such as pulmonary hypertension, congestive heart failure, and influenza/pneumonia. COPD, a comorbidity sharing common risk factors (e.g., smoking) and contributing to the decline in respiratory function, was selected as the UCOD in 3.9% but mentioned in as many as 12.9% of the death certificates with IPF. Gastro-oesophageal reflux disease has been proposed as a risk factor for IPF through chronic lung damage due to acid microaspiration [22,23]. Our study showed an association between this condition and IPF (OR: 2.9, 95% CI: 1.3, 6.5), despite the fact that esophagitis or reflux disease were only reported in 0.3% of death certificates with a mention of IPF.

Patients with interstitial lung disease, including IPF, are at an increased risk of death from COVID-19, especially those with poor lung function and obesity [24]. In a cohort study carried out in a large number of general practices in England, subjects affected by IPF displayed a modestly increased risk of COVID-19 related hospitalization and death [25]. To date, few population-based studies are available on the causes of increased mortality during the COVID-19 pandemic; such reports are limited to broad nosological categories or common diseases and rely on the UCOD [26,27]. The present analysis, based on MCOD data, highlights an increased mortality associated with IPF during the first epidemic wave in northern Italy, therefore supporting the current strategies prioritizing COVID-19 vaccination of subjects affected by IPF (Gazzetta Ufficiale della Repubblica Italiana, 24 marzo 2021, Serie 72°, p. 42).

The main limitation of the study was the lack of validation of IPF reporting in death certificates. A study of IPF using a combination of hospital discharge and mortality records found a remarkable degree of underdiagnosis of IPF in a large administrative region in central Italy [20]. While it is unlikely that a death certifier would mention a diagnosis of IPF without some confirmation from secondary care, some cases may have been missed due to diagnostic transfer to other disease entities within pulmonary fibrosis that are difficult to differentiate clinically [19]. A limited proportion of death certificates mentioned both pulmonary fibrosis and other casually related conditions; however, time trends observed in the sensitivity analysis excluding such deaths were consistent with the main analyses.

## 5. Conclusions

Our data suggest that, during the last decade, mortality for IPF has been increasing in the elderly male population of a large area in northeastern Italy, but not among women. The burden due to IPF is heavier than what would be expected by routine mortality statistics, since less than two out of three deaths with a mention of IPF are directly attributed to this condition. The COVID-19 epidemic was accompanied by an increase in IPF-related mortality in northeastern Italy.

## Figures and Tables

**Figure 1 ijerph-18-07249-f001:**
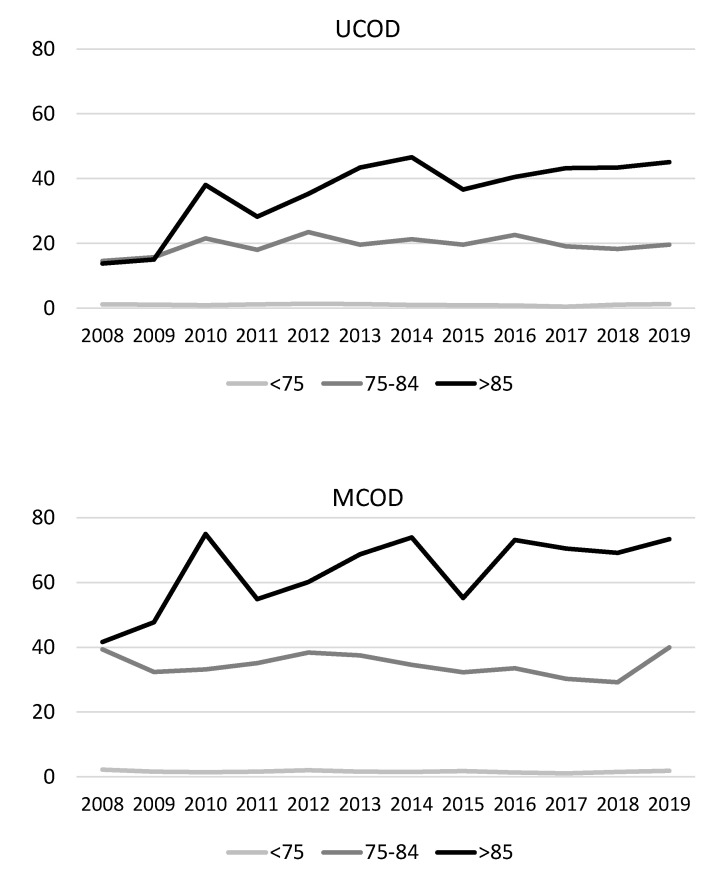
Time trends in age-standardized mortality rates for IPF among males by age class, assessed by UCOD and MCOD analyses. Veneto region, 2008–2019.

**Figure 2 ijerph-18-07249-f002:**
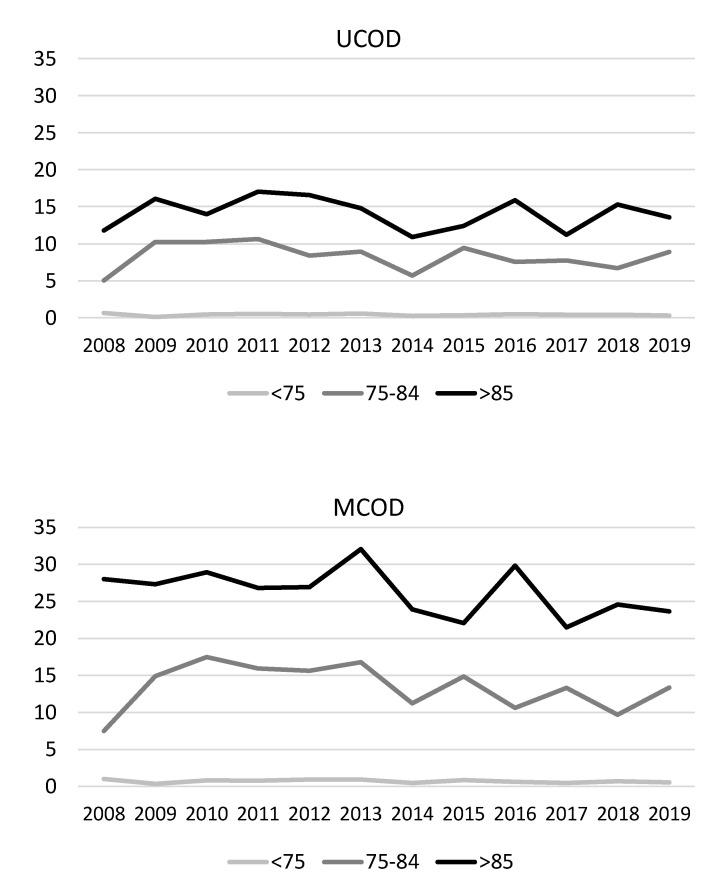
Time trends in age-standardized mortality rates for IPF among females by age class, assessed by UCOD and MCOD analyses. Veneto region, 2008–2019.

**Figure 3 ijerph-18-07249-f003:**
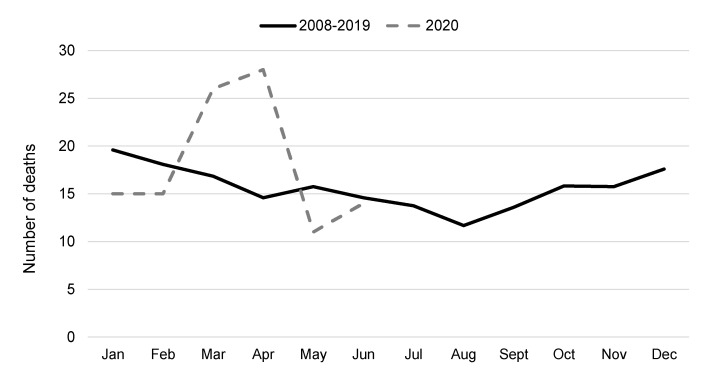
Number of death certificates with any mention of IPF by month: average count in 2008–2019 and provisional data from the first semester 2020.

**Table 1 ijerph-18-07249-t001:** Number of deaths, proportional mortality (% of all deaths), and mortality rates for IPF as the underlying cause of death (UCOD) or mentioned in any position on death certificates (MCOD). Veneto region, 2008–2019.

		Number of Deaths		Proportional Mortality		Rate per 100,000/Year	
	Age Class	UCOD	MCOD	UCOD	MCOD	UCOD	MCOD
Males	<45	7	11	0.1%	0.1%	0.0	0.1
	45–49	2	3	0.0%	0.1%	0.1	0.1
	50–54	7	12	0.1%	0.2%	0.3	0.5
	55–59	13	23	0.1%	0.3%	0.7	1.2
	60–64	42	64	0.3%	0.5%	2.4	3.7
	65–69	62	100	0.3%	0.5%	3.9	6.3
	70–74	123	202	0.4%	0.7%	9.0	14.7
	75–79	176	295	0.4%	0.7%	16.3	27.2
	80–84	177	331	0.4%	0.7%	24.7	46.2
	85–89	146	238	0.3%	0.5%	40.0	65.2
	90+	45	87	0.1%	0.2%	33.0	63.9
	Overall	800	1366	0.3%	0.5%	2.8	4.8
Females	<45	4	11	0.1%	0.3%	0.0	0.1
	45–49	2	5	0.1%	0.2%	0.1	0.2
	50–54	9	12	0.2%	0.3%	0.4	0.5
	55–59	13	18	0.3%	0.4%	0.7	0.9
	60–64	11	25	0.1%	0.3%	0.6	1.4
	65–69	22	37	0.2%	0.3%	1.3	2.2
	70–74	54	92	0.3%	0.5%	3.4	5.8
	75–79	101	160	0.4%	0.6%	7.1	11.3
	80–84	117	194	0.2%	0.4%	10.1	16.7
	85–89	117	226	0.2%	0.3%	14.5	28.1
	90+	60	105	0.1%	0.1%	13.2	23.1
	Overall	510	885	0.2%	0.3%	1.7	3.0

**Table 2 ijerph-18-07249-t002:** Distribution of the underlying causes of death in 2251 certificates with any mention of IPF.

Condition (ICD10)	Number of Deaths			% of All Deaths		
	Overall	Males	Females	Overall	Males	Females
IPF (J84.1)	1310	800	510	58.2%	58.6%	57.6%
COPD (J40–J44, J47)	87	62	25	3.9%	4.5%	2.8%
Other respiratory diseases (J00–J99)	50	28	22	2.2%	2.0%	2.5%
Lung cancer (C33–C34)	87	71	16	3.9%	5.2%	1.8%
Other neoplasms (C00–D48)	103	67	36	4.6%	4.9%	4.1%
Ischemic heart diseases (I20–I25)	149	106	43	6.6%	7.8%	4.9%
Other circulatory diseases (I00–I99)	187	110	77	8.3%	8.1%	8.7%
Digestive diseases (K00–K99)	30	14	16	1.3%	1.0%	1.8%
Conditions causally related to pulmonary fibrosis (M05–M08, M32–M36, D86, J60–J65, J67, J70.1)	104	35	69	4.6%	2.6%	7.8%
Other diseases	144	73	71	6.4%	5.3%	8.0%
Overall	2251	1366	885	100%	100%	100%

**Table 3 ijerph-18-07249-t003:** Proportion of death certificates mentioning selected complications/comorbidities and associations between IPF and these conditions.

Condition (ICD10)	Deaths with IPF (*n* = 2251)	Deaths without IPF (*n* = 555,681)	Odds Ratio (95% Confidence Interval)
Pulmonary hypertension (I27)	12.5%	0.8%	15.7 (13.8–17.9)
Congestive heart failure (I50.0)	2.5%	1.8%	1.5 (1.2–2.0)
Influenza, pneumonia (J09–J18)	15.3%	10.2%	1.6 (1.5–1.8)
Pulmonary embolism (I26.x)	2.8%	2.0%	1.4 (1.0–1.7)
COPD (J40–J44, J47)	12.9%	7.2%	1.8 (1.6–2.0)
Esophagitis/reflux (K20, K21)	0.3%	0.1%	2.9 (1.3–6.5)
Conditions causally related to pulmonary fibrosis (M05–M08, M32–M36, D86, J60–J65, J67, J70.1)	8.0%	0.8%	9.8 (8.3–11.5)

## Data Availability

Restrictions apply to the availability of these data. Data were obtained from Azienda Zero Regione del Veneto and are available from the authors with their permission.

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
