# Peer review of "Mortality Associated with Idiopathic Pulmonary Fibrosis in Northeastern Italy, 2008–2020: A Multiple Cause of Death Analysis"

_ijerph, 2021, doi:10.3390/ijerph18147249_

Round 1

Reviewer 1 Report

This is a very interesting paper on mortality from IPF in the Veneto Region in 2008-2020 using MCOD. I feel that this is important research, although mostly descriptive it is well-conducted and well-written. However, I have a set of comments and considerations that may help the authors improving the paper:

  • I would add some additional information about IPF as not all readers may be familiar with it. A general overview on prevalence or lethality will be good to know
  • Aims: Nothing is mentioned about the underlying cause-of-death analyses on deaths with IPF listed in the death certificate
  • In table 3 OR of the proportion of certificates mentioning selected complications/causes of death in deaths with and without IPF are estimated. However, this is only explained in methods in one sentence. Could the authors provide more information on the conditional logistic regression to make the results more transparent?
  • Figure 1. Given that the sample of deaths with IPF is not that big (in absolute terms, and as compared to other major diseases) I would like to see the 95%CI of rates. Could it be that the “increase” has only weak evidence supporting it?
  • Figure 2. Again, it could be that the observed lines contain a lot of variability therein. That is, it is clear from the graph that there was an increase of deaths with mention of IPF, but what did it happen in INDIVIDUAL previous years (2008, 2009….)? It would be nice to have this information, at least as supplementary material.
  • First paragraph. The authors mention a “sharp increase”. However, looking at figure 1 I can see that the rates for 2007 and 2008 were very low. If we exclude these two points of data the increase does not seem very strong. Could the authors comment on that? Were, for any reason, 2007 and 2008 particular years in relation to mortality coding / respiratory disease /IPF?
  • Given my previous comments, I would be a bit more careful in using some specific words in the discussion. For example, I would use “suggests” instead of “demonstrates” when discussing the increased mortality associated with IPF in the first wave of covid-19
  • Last, but not least, I wondered whether the authors did not choose to focus on a larger group of causes (e.g. J80-J84, or COPD).

Reviewer 2 Report

The article addresses a relevant topic for Public Health and expands the knowledge about mortality from IFP, presenting data for Italy and considering two approaches, based on UCOD and MCOD.

The main concern is the time series analysis, which can be improved.

It is unclear why temporal trends were assessed in only two age groups (<85 and ≥85 years). If there is an upward trend in the ≥85 age group, would it not be reasonable to consider that the same trend could be verified in the 80-84 or 75-79 age groups?

I believe that to achieve the proposed goals it would be necessary to assess trends in each age group. Maybe it would be interesting to build age groups every 10 years, so as to have more cases in each group.

I believe that the Joinpoint software assesses trends by adjusting a model for each age group separately, which is not ideal. Furthermore, this software does not present a residual analysis for the fitted models, which can lead to inappropriate modeling and errors in the AAPC estimates.

Thus, I suggest evaluating the trend in all age groups using a single regression model, possibly Poisson regression or regression with a negative binomial response, whichever is more appropriate.

It would be necessary to present graphs with rates over time for all age groups considered. For visualization purposes, I suggest 4 graphics: M-UCOD, M-MCOD, F-UCOD, F-MCOD.

Some additional comments:

In table 2, add a line with the totals at the end of the table.

In Table 3, I believe that the statistic presented is the prevalence ratio and not the odds ratio.

In row 129, mortality rates for Italy (2008-2020) and European Union (2001-2013) are compared. As rates are increasing in most EU countries, it would be interesting to compare estimates for similar periods.

In line 149, it is not clear how the excess deaths were calculated.

Round 2

Reviewer 2 Report

There was a great improvement in the article and I would like to congratulate the authors. The study appears to have been very well conducted, is well written, and makes contributions to the understanding of mortality due to IPF.

I have few considerations to make:

Line 95: The AAPC in age-standardized rates can be obtained for both sexes by a single Poisson model with the explanatory variables calendar year, sex, and their respective interaction. In fact, it makes no sense to use jointpont in this study, for reasons already mentioned in the first review.

lines 168 to 172: See previous comment.

Line 110: I believe IPF is the dependent variable.

line 165: ... "among females, on average, wich are both ..." instead of "among females, wich are both ..."

Line 181: Report that no trends were found for women.

Regards,

The reviewer.
